# D³U-Net: Dual-Domain Collaborative Optimization Deep Unfolding Network for Image Compressive Sensing

Kai Han
Beijing University of Technology
Beijing, China
hankai@emails.bjut.edu.cn

Jin Wang*
Beijing University of Technology
Beijing, China
ijinwang@bjut.edu.cn

Yunhui Shi
Beijing University of Technology
Beijing, China
syhzm@bjut.edu.cn

Nam Ling
Santa Clara University
Santa Clara, USA
nling@scu.edu.cn

Baocai Yin
Beijing University of Technology
Beijing, China
ybc@bjut.edu.cn

## Abstract

Deep unfolding network (DUN) is a powerful technique for image compressive sensing that bridges the gap between optimization methods and deep networks. However, DUNs usually rely heavily on single-domain information, overlooking the inter-domain dependencies. Therefore, such DUNs often face the following challenges: 1) information loss due to the inefficient representation within a single domain, and 2) limited robustness due to the absence of inter-domain dependencies. To overcome these challenges, we propose a deep unfolding framework D³U-Net that establishes a dual-domain collaborative optimization scheme. This framework introduces both visual representations from the image domain and multi-resolution analysis provided by the wavelet domain. Such dual-domain representations constrain the feasible region within the solution space more accurately. Specifically, we design a consistency-difference collaborative mechanism to capture inter-domain dependencies effectively. This mechanism not only enhances the fidelity of reconstruction but also enriches the depth and breadth of extracted features, improving the overall robustness and reconstruction quality. Moreover, we develop an inter-stage transmission pathway to minimize the information loss during transmission while broadcasting multi-scale features in a frequency-adaptive manner. Extensive experimental results on various benchmark datasets show the superior performance of our method.

## CCS Concepts

• **Computing methodologies** → **Image processing**; **Reconstruction**.

## Keywords

Compressed sensing; deep unfolding; dual-domain collaboration

*Jin Wang is the corresponding author.

**ACM Reference Format:**
Kai Han, Jin Wang, Yunhui Shi, Nam Ling, and Baocai Yin. 2024. D³U-Net: Dual-Domain Collaborative Optimization Deep Unfolding Network for Image Compressive Sensing. In *Proceedings of the 32nd ACM International Conference on Multimedia (MM '24), October 28–November 1, 2024, Melbourne, VIC, Australia.* ACM, New York, NY, USA, 9 pages. https://doi.org/10.1145/3664647.3681532

## 1 Introduction

Compressive sensing (CS) [11] is a promising methodology, which can reconstruct signals $x \in \mathbb{R}^N$ exactly from much fewer measurements $y \in \mathbb{R}^M$ than the requirement of classical Nyquist theory. Here, $y = Ax, M \ll N, A \in \mathbb{R}^{M \times N}$ is the sampling matrix. Therefore, CS is widely used in applications such as remote sensing, magnetic resonance imaging (MRI), snapshot compressive imaging, and radar imaging, where data acquisition is costly or time-consuming.

CS is a typical ill-posed problem in signal processing, with numerous approximate solutions, presenting a significant challenge to reconstruct the original signal accurately. Over the past decades, researchers have devised a multitude of algorithms aimed at integrating rich prior knowledge into the CS framework to address this challenge, such as structural sparsity in some transformation domains [5], non-local self-similarity [23], total variation [22], and low rank [45]. Afterwards, many nonlinear iterative methods are developed, such as orthogonal matching pursuit [27], greedy matching pursuit algorithm [24], gradient descent algorithm [44], convex optimization algorithm [7], and so on. Despite the benefits of robust convergence and solid theoretical bias, these methods are frequently burdened by high computational demands.

Recently, deep learning-based methods [14, 29, 30] have achieved remarkable success, owing to their powerful learning ability, which enables them to extract robust priors from extensive datasets. These methods can be divided into two primary groups: deep black box network (DBN) and deep unfolding network (DUN). DBNs [12, 36, 37] can learn a direct deep inverse mapping from the measurement domain to the original image domain by end-to-end networks. DBNs have been widely employed in early deep learning-based studies due to their simplicity and effectiveness. However, DBNs are trained as black boxes and lack interpretability, significantly limiting the further improvement of reconstruction quality. Thus, DUNs [15, 21, 46] are proposed with great interpretability and impressive performance. DUNs usually unfold optimization methods

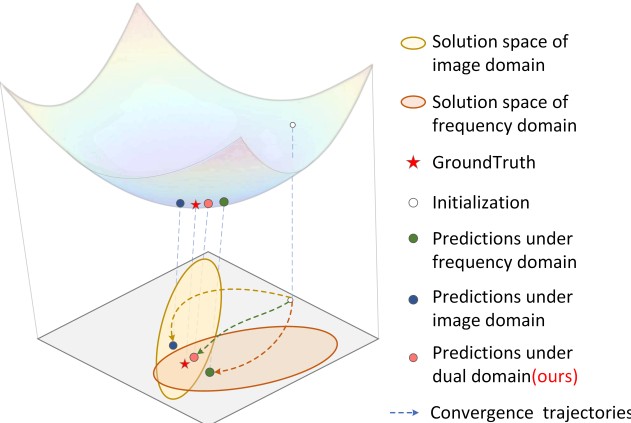

**Figure 1: Convergence trajectories of reconstruction under different conditions.**

into trainable networks in the image domain, such as the proximal gradient descent methods [6, 10, 42, 46], approximate message passing methods [3, 25, 38], and alternating direction methods of multipliers[39]. However, most existing DUNs are designed based on traditional single-domain unfolding, in which each stage processes single-domain information as input and output, resulting in limited representation capacity and robustness.

More recent research [8, 35, 48] has exploited the multi-domain information for signal recovery that incorporates image prior knowledge with frequency information. However, most of them cannot efficiently coordinate the consistency and differences across different domains. In summary, most existing CS methods often face the following challenges. *1) Information loss due to the inefficient representation within single-domain.* Within each stage, single-domain information is often used as a single-channel image to bridge intermodule communication, resulting in limited representation capacity of networks, posing challenges in fully capturing the complex features and structures of images. *2) limited robustness due to the absence of inter-domain dependencies.* Merely relying on simple addition or concatenation for cross-domain collaboration often results in suboptimal utilization or introducing noise. Neglecting the consistency and difference among multiple domains can destroy their correlation, leading to misinterpretation of images and inaccurate restoration of subtle features.

To overcome the aforementioned challenges, we propose a novel dual-domain cooperative optimization framework called $D^3$U-Net for image CS, which aims to exploit the unique characteristics of different domains. The robustness of our method against noise is augmented during the reconstruction process, mitigating artifacts and information loss by applying the proposed consistency-difference collaborative mechanism. By introducing features from both domains, our framework can accurately represent the intricate structure and texture, enhancing robustness and minimizing artifacts. Fig. 1 provides the assumed convergence paths under various conditions. The main contributions are summarized as follows:

- A novel dual-domain collaborative optimization framework named $D^3$U-Net is proposed for image CS, where both visual representation from the image domain and multi-resolution

analysis from the wavelet domain are utilized to more accurately constrain the feasible solution space, breaking the limitation of inefficient representation within single-domain.
- A consistency-difference collaborative mechanism is designed to capture inter-domain dependencies. This mechanism not only takes full advantage of consistency to guide cross-domain fusion but also explores differences to facilitate information compensation.
- An inter-stage transmission path is presented to efficiently broadcast the multi-scale features in a frequency-adaptive manner, which can effectively mitigate the intrinsic information loss.
- Extensive experiments demonstrate that our proposed method achieves excellent performance.

## 2 Related Works

### 2.1 Deep Black Box Networks

Deep black box networks [6, 20, 26, 30, 41] are designed to establish a learnable mapping from the measurement domain to the original image domain, enabling accurate reconstruction from CS measurements. Block-by-block CS methods have been widely studied in the early stage due to their simplicity and effectiveness. However, these block-by-block methods often suffer from noticeable block artifacts. To address this issue, several methods [9, 17] have been proposed to leverage deep image priors in the whole image space. Subsequently, efficient functional modules have been integrated into CS frameworks to further enhance the reconstruction performance, such as self-attention mechanisms[13, 19, 28], multi-scale feature fusion techniques [4], scalable sampling methods [29], and so on. However, DBNs are usually trained as black boxes, lacking a solid theoretical foundation and explainability. This limitation can restrict the reliability and controllability of low-level vision tasks.

### 2.2 Deep unfolding Networks

Deep unfolding networks [31–33, 43, 50], taking advantage of optimization-based algorithms and deep learning techniques, have been garnering growing interest in the field of low-level computer vision tasks. Researchers in this field have proposed many methods that incorporate CNN-based denoisers with various optimization algorithms (ISTA [46], ADMM [39], AMP [49], HQS [1], and so on). Mathematically, DUN for the CS reconstruction task is usually formulated as the bi-level optimization problem:

$$\begin{cases} \min_{\Theta} \sum_{j=1}^{N_a} \mathcal{L}(\hat{x}_j, x_j; \Theta), \\ \text{s.t. } \hat{x}_j = \arg\min_{x} \frac{1}{2}||Ax - y||_2^2 + \lambda\Psi(x), \end{cases} \quad (1)$$

where $\{(\hat{x}_j, x_j)\}_{j=1}^{N_a}$ is the given data pairs in training set, $A$ denotes the sampling matrix, $y$ indicates the CS measurements, $\Theta$ is the learnable parameter of DUN and $\lambda$ is used to control the contribution of the regularization/prior term. $\mathcal{L}(\cdot)$ means the loss function of a specific DUN. However, most existing DUNs only consider image-domain based mapping. Although some DUNs have integrated intermediate features as auxiliary information into cross-stage communication, the fundamental concept of image-domain

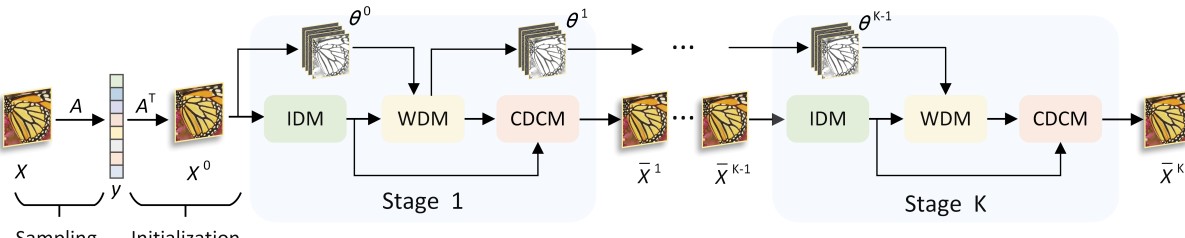

**Figure 2: The structure of the proposed dual-domain framework. IDM means the image domain mapping module. WDM indicates the wavelet domain mapping module. CDCM denotes the consistency-difference collaborative mechanism.**

based unfolding has not changed, which limits further performance improvement.

## 3 Proposed Method

### 3.1 Theoretical Basis of Our Dual-domain Model

As shown in Fig. 2, our dual-domain framework is an end-to-end deep unfolding network, which consists of three parts: sampling, initialization, and reconstruction. The sampling and initialization process can be formulated as follows:

$$y = Ax_{\text{gt}}, \tag{2}$$

$$x_{\text{init}} = A^T y, \tag{3}$$

where $y$ is the CS measurements, $A$ means the sampling matrix, $x_{\text{gt}}$ denotes the input image, $x_{\text{init}}$ represents the initial reconstruction. Eq. (2) and Eq. (3) indicate the sampling and initialization process, respectively.

The deep reconstruction process is performed on the initial reconstruction result $x_{\text{init}}$ and improves its quality. We divide the reconstruction model into $K$ stages. Each stage alternatively implements the projection in both the image domain and the wavelet domain. The main idea leads to the integration of image-domain and wavelet-domain priors can be formulated as follows:

$$\underset{\theta, x}{\text{argmin}} \ \frac{1}{2}\|y - Ax\|_2^2 + \frac{\lambda}{2}\|x - \Phi^{-1}\theta\|_2^2 + \alpha\mathcal{H}(\theta) + \beta\mathcal{F}(x), \tag{4}$$

where $y$ is the CS measurements, $A$ is the sampling matrix, $\Phi^{-1}$ denotes the inverse wavelet transform. $\mathcal{H}(\cdot)$ and $\mathcal{F}(\cdot)$ are the prior terms of wavelet-domain and image-domain, respectively. $\alpha, \lambda, \beta$ are the balancing coefficients. The variables $x$ and $\theta$ represent the reconstructions in the image domain and wavelet domain, respectively. To simplify the solving process, we decompose the above optimization problem into two sub-problems: the $\theta$-subproblem and the $x$-subproblem, which are delineated as follows.

**Image-domain subproblem:** The $x$-subproblem is equivalent to solving the optimization problem as follows:

$$x^k = \underset{x}{\arg\min} \ \frac{1}{2}\|y - Ax\|_2^2 + \frac{\lambda}{2}\|x - \Phi^{-1}\theta\|_2^2 + \beta\mathcal{F}(x). \tag{5}$$

The mapping process for solving the corresponding optimization $x$-subproblem can be described as follows:

$$r^k = x^{k-1} - A^T(Ax^{k-1} - y) - \lambda(x^{k-1} - \Phi^{-1}\theta^{k-1}), \tag{6}$$

$$x^k = \underset{x}{\arg\min} \ \|x - r^k\|_2^2 + \beta\mathcal{F}(x) = \mathcal{D}(r^k), \tag{7}$$

where $\mathcal{D}(\cdot)$ denotes a denoising mapping network. The Eq. (5) can be transformed into Eq. (7) to facilitate the solution, by leveraging the gradient descent process in Eq. (6). The structural detail of $\mathcal{D}(\cdot)$ is shown in Fig. 3. We use the denoising block $\mathcal{N}_k(\cdot)$ to remove the noise of $x^{k-1}$ to get the $k$-th reconstruction result $x^k$ in the image domain. Each denoising module consists of four $3 \times 3$ Conv layers. There is a ReLU activation function between adjacent Conv layers.

**Wavelet-domain subproblem:** The $\theta$-subproblem in Eq. (4) can be formulted as follows:

$$\theta^k = \underset{\theta}{\arg\min} \ \frac{\lambda}{2}\|x^k - \Phi^{-1}\theta\|_2^2 + \alpha\mathcal{H}(\theta),$$

$$= \underset{\theta}{\arg\min} \ \frac{\lambda}{2}\|\theta_{x^k} - \theta\|_2^2 + \alpha\mathcal{H}(\theta). \tag{8}$$

Here, $\theta_{x^k}$ is the wavelet coefficients, which is obtained by applying a wavelet decomposition to the image-domain reconstruction $x^k$. We employ a prior-term solving module (PTSM) to solve the problem in Eq. (8), which can be written as follows:

$$\theta^k = \mathcal{P}(x^k, \theta^{k-1}, \Phi^{-1}, \lambda, \alpha), \tag{9}$$

where $\mathcal{P}(\cdot)$ represents the PTSM. Moreover, we introduce the wavelet domain information $\theta^{k-1}$ from the previous stage to leverage additional prior information (i.e., the underlying structure and sparsity characteristics in the wavelet domain.). It can help overcome the constraints imposed by single-stage transmission, enhancing the overall reconstruction process by exploiting the knowledge gained from previous stages. As shown in Fig. 3, the PTSM comprises a multi-scale network, which can explore the feature representations at different scales to improve model performance and generalization.

### 3.2 Consistency-Difference Collaboration

To ensure that the dual-domain priors can effectively guide the target image reconstruction, we need to facilitate the collaboration of dual-domain features. This is achieved by exploiting the consistency and difference between the two domains, enabling the sufficient harnessing of complementary information from both perspectives. In this paper, we propose a consistency-difference collaboration

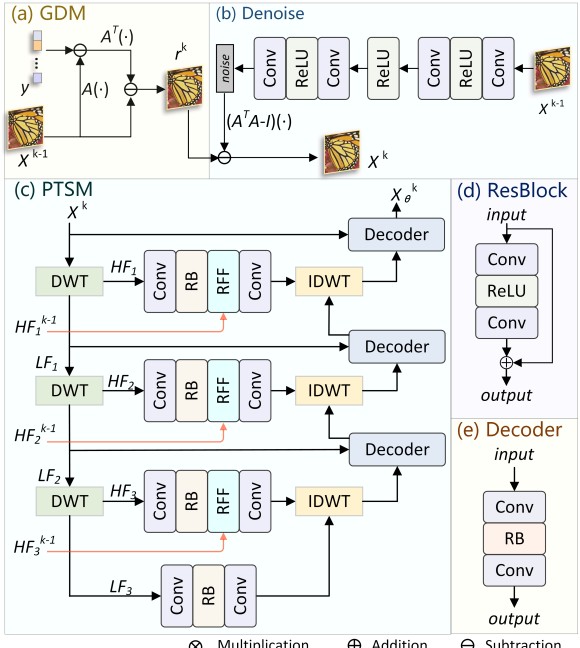

**Figure 3: The detail of different blocks. The IDM consists of two key components: (a) GDM (gradient descent sub-module) and (b) denoise sub-module. As shown in sub-figure (c), the WDM is composed of the PTSM (Prior-term Solving Module).**

mechanism (CDCM) to realize information communication from different domains, as shown in Fig. 4.

**Consistency:** The consistency learning process can be formulated as follows:

$$F_x = \mathcal{B}_{\text{consistency}}(x^k), \tag{10}$$

$$F_\theta = \mathcal{B}_{\text{consistency}}(\theta^k), \tag{11}$$

$$F_C = \mathcal{R}_C(F_x \odot F_\theta \odot x_{\text{init}}), \tag{12}$$

where $\mathcal{B}_{\text{consistency}}(\cdot)$ denotes the cross-domain consistency mapping module. $\mathcal{B}_{\text{consistency}}(\cdot)$ consists of a $1 \times 1$ Conv and a ResBlock. The dual domains share weights of $\mathcal{B}_{\text{consistency}}(\cdot)$. $F_C$ indicates the cross-domain consistency. $F_x$ and $F_\theta$ are the output of consistency mapping module. $\mathcal{R}_C(\cdot)$ is a ResBlock, as shown in Fig. 3.

$$C_x = \text{Conv}_{1\times 1}(F_C + F_x), \tag{13}$$

$$C_\theta = \text{Conv}_{1\times 1}(F_C + F_\theta), \tag{14}$$

$$C_A = C_x + C_\theta, \tag{15}$$

where $\text{Conv}_{1\times 1}(\cdot)$ means the $1 \times 1$ convolution operation, $\odot$ is the element-wise multiplication, $C_A$ contains augmented consistency of features and rich details from the dual-domain space. $C_x$ and $C_\theta$ are the consistency extracted from the image domain and wavelet domain, respectively.

**Difference:** The difference learning process can be formulated as follows:

$$F_D = \mathcal{R}_D(|x - IDWT(\theta)| \odot x_{init}), \tag{16}$$

where $\mathcal{R}_D(\cdot)$ is a ResBlock, as shown in Fig. 3. And $|\cdot|$ is the absolute value operation. $\odot$ means the element-wise multiplication.

$F_D$ denotes the difference derived from both the image domain and the wavelet domain.

**Dual-domain collaboration:** We utilize a residual transformer architecture to collaborate the dual-domain information as shown in Fig. 4. The transformer utilizes self-attention to weigh the importance of different parts within the input. The keys $K$ and values $V$ represent different aspects of the data that the network should focus on. By using the consistent information derived from both domains as the keys and values, the transformer can effectively learn relationships and dependencies across the wavelet domain and the image domain, enhancing the fusion of dual-domain information. The queries $Q_x$ and $Q_\theta$ come from the dual domain.

$$Q_x = \text{DWConv}(x), \tag{17}$$

$$Q_\theta = \text{DWConv}(\theta), \tag{18}$$

$$K = \text{DWConv}(C_A), \tag{19}$$

$$V = \text{DWConv}(C_A). \tag{20}$$

The consistent information serves as guidance for the attention mechanism, ensuring that the transformer model attends to the most relevant parts of the data from both domains during the fusion process. Then, we compute two similarity maps $M_x$ and $M_\theta$.

$$M_x = \text{Softmax}(K^T Q_x + M_x^{\text{pre}}), \tag{21}$$

$$M_\theta = \text{Softmax}(K^T Q_\theta + M_\theta^{\text{pre}}), \tag{22}$$

where $\text{Softmax}(\cdot)$ is the softmax activation function. $M_x^{\text{pre}}$ and $M_\theta^{\text{pre}}$ denote similarity maps from the previous stage. This approach can lead to a more robust and accurate representation of the data, as it incorporates insights from both the wavelet and image domains. The process can be expressed as follows:

$$x_{\text{fuse}} = \text{concat}(\text{Conv}(V^T M_x), \text{Conv}(V^T M_\theta)) \tag{23}$$

where $\text{concat}(\cdot)$ denotes the operation of concatenating features by channel. $\text{Conv}(\cdot)$ indicates a $1 \times 1$ Conv layer.

Finally, the Feed-Forward Block (FFB) is a key element that boosts the representation ability of the model through nonlinear transformations and feature interactions, ensuring dimensional integrity and training robustness with residual connections. The process is shown as follows:

$$\overline{x}_{\text{out}} = \text{FFB}(x_{\text{fuse}}, F_D), \tag{24}$$

where $\overline{x}_{\text{out}}$ is the output of current stage.

## 3.3 Informative Inter-stage Transmission

The PTSM incorporates informative inter-stage transmission for wavelet domain optimization. The decomposition of an image into multi-scale subbands facilitates the retention of multiple levels of detail and structure. As shown in Fig. 3, we use a multi-scale architecture to map the wavelet domain optimization process, the intermediate features of each level can be transmitted to the next stage to reduce the information loss. For each scale, the encoder is composed of two Conv layers, one ResBlock, and one RFF block. The decoder consists of two $1 \times 1$ Conv layers and a ResBlock between them. We use the RFF block to further refine these intermediate features to provide more details for the next stage and the larger scale. The RFF block denotes channel attention, which can extract key components from the previous stage and the next scale.

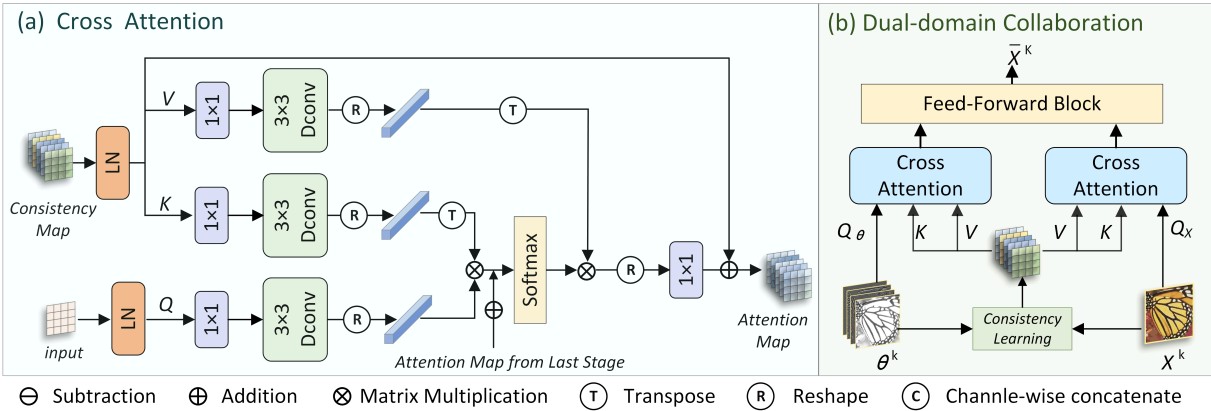

**Figure 4: The detail of CDCM. It consists of two key components: the cross-attention and the dual-domain collaboration blocks.**

## 3.4 Loss Function

The loss function can be divided into three parts: the MSE loss, the texture loss, and the consistency loss. The computing process can be formulated as follows:

$$\mathcal{L}_1 = \frac{1}{N_a N_b}||\hat{x} - x||, \tag{25}$$

$$\mathcal{L}_2 = \max(\eta||\hat{\theta}_H|| - ||\theta_H||, 0), \tag{26}$$

$$\mathcal{L}_3 = ||x_\theta - x_{im}|| + ||x_\theta - x|| + ||x - x_{im}||, \tag{27}$$

$$\mathcal{L}_{total} = \mathcal{L}_1 + 0.001 * \mathcal{L}_2 + 0.0001 * \sum_k e^k * \mathcal{L}_3^k, \tag{28}$$

where $\mathcal{L}_1, \mathcal{L}_2, \mathcal{L}_3$ are the MSE loss, the texture loss, and the consistency loss, respectively. $N_a$ and $N_b$ are the total number of the training set and the size of images within the training set, respectively. $\mathcal{L}_{total}$ indicates the total loss of our model. $\hat{x}$ denotes the finally reconstruction. $x$ is the ground truth. $\hat{\theta}_H$ is the high frequency subbands of wavelet domain reconstruction. $\theta_H$ is the high frequency subbands of ground truth. $x_\theta$ and $x_{im}$ mean the reconstruction from the wavelet domain and image domain, respectively. As the number of iterations increases, the discrepancy between the reconstructions in the wavelet domain and the image domain diminishes gradually, strengthening the consistency of the information across both domains. The coefficient of $\mathcal{L}_c$ increases with the number of iterations. Thus, we set $e^k$ to control the $\mathcal{L}_c$. $*$ denotes multiplication.

## 4 Experiment

### 4.1 Implementation and Training Details

We select 800 images from the coco dataset for training. The training images are cropped to about 200000 patches of $64 \times 64$ pixel size, which are randomly extracted from images. We use PyTorch 1.7 and Python 3.7 and train our model by exploiting the Adam optimizer on an NVIDIA RTX 3090 GPU. All models are trained for 150 epochs with batch size 32 and learning rate $1 \times 10^{-4}$. Before training, the sampling matrix $A$ is initialized as a random Gaussian matrix. The CS reconstruction accuracy on all datasets is evaluated with the Peak Signal to Noise Ratio (PSNR) and Structural Similarity (SSIM). Set5 [2], Set11 [18], and Urban100 [16] are used test datasets, which

are widely used to evaluate the performance of various low-level vision tasks.

### 4.2 Comparisons with Other Methods

Apart from some experimental results provided by the authors, the results of the other comparison methods are retrained in the same environment as our model. Their source codes are officially published by their authors. Moreover, additional experimental results are presented in the supplementary material. Table 1 and Table 2 show the average PSNR/SSIM results of our model and previous state-of-the-art methods. For example, our method outperforms AMP-Net [49], COAST-Net [43], ISTA-Net++ [42], MADUN [31], BCS-Net [50], DPUNet [40], DPC-DUN [32], SODAS-Net [33] by 1.32 dB, 2.09 dB, 4.52 dB, 2.44 dB, 1.49 dB, 1.42 dB, 1.48 dB, 1.38 dB, and 1.94 dB in terms of PSNR on Set11 dataset when the CS sampling ratio is 10%, respectively. In addition, the average SSIM gain of our method over these comparison methods is 0.0193, 0.0367, 0.1024, 0.0454, 0.0217, 0.0335, 0.0017, 0.0187, and 0.0032, respectively. Fig. 6 and Fig. 5 further show the visual comparisons on challenging images at a 10% CS sampling ratio, which can be seen that our method can recover much clearer edge information than other methods.

### 4.3 Ablation Studies

**Impact of Dual-domain information.** We conduct extensive ablation experiments on dual-main priors at 10% CS sampling ratio on the Urban100 dataset, as shown in Table 3. Case (c) achieves 0.34 dB and 0.39 dB improvement compared with Case (a) and Case (b) in terms of PSNR, which validates the superiority of our idea to cooperate with the dual-domain information.

**Impact of Consistency-difference.** As shown in Table 4, we explore the impact of consistency and difference at 10% CS sampling ratio on Urban100. Case (c) attains a PSNR improvement of 0.40 dB and 0.28 dB over Case (a) and Case (b), respectively, demonstrating the effectiveness of leveraging consistency and difference.

**Impact of Inter-Stage Transmission of Multi-scale Features.** We analyze the effect of inter-stage transmission of multi-scale features at 10% CS sampling ratio in Table 5. Case (b) achieves 0.16 dB improvement compared with Case (a) on Urban100, which validates

**Table 1: Average PSNR(dB)/SSIM comparisons on Urban100. The best and second-best results are in bold and underlined, respectively.**

| Dataset | Method | Sampling Rate | | | | |
|---------|--------|------|------|------|------|------|
| | | 10% | 20% | 30% | 40% | 50% |
| Urban100 | DPA-Net [34] | 24.55/0.7841 | -/- | 29.47/0.9034 | 31.09/0.9311 | 32.08/0.9447 |
| | AMP-Net [49] | 25.96/0.8133 | 29.50/0.8974 | 32.07/0.9352 | 34.22/0.9569 | 36.16/0.9706 |
| | COAST [43] | 25.94/0.8038 | 29.70/0.8940 | 32.20/0.9317 | 34.21/0.9528 | 35.99/0.9665 |
| | ISTA-Net [46] | 23.28/0.7094 | 26.90/0.8364 | 29.62/0.8980 | 31.87/0.9322 | 33.98/0.9538 |
| | ISTA-Net++ [42] | 24.78/0.7607 | 28.55/0.8687 | 31.08/0.9152 | 33.10/0.9402 | 34.86/0.9560 |
| | OPINE-Net [47] | 26.56/0.8345 | 30.07/0.9088 | 32.64/0.9419 | 34.66/0.9600 | 36.64/0.9725 |
| | DPUNet [40] | 26.10/0.8226 | 29.71/0.9027 | 32.23/0.9378 | 34.30/0.9573 | 36.10/0.9693 |
| | DPC-DUN [32] | 26.96/0.8361 | -/- | 33.53/0.9449 | 35.61/0.9624 | 37.52/0.9737 |
| | SODAS-Net [33] | 26.22/0.8055 | 29.51/0.8950 | 33.15/0.9412 | 35.27/0.9599 | 37.14/0.9721 |
| | **Ours** | **28.01/0.8611** | **31.67/0.9248** | **34.51/0.9544** | **36.53/0.9688** | **38.66/0.9790** |

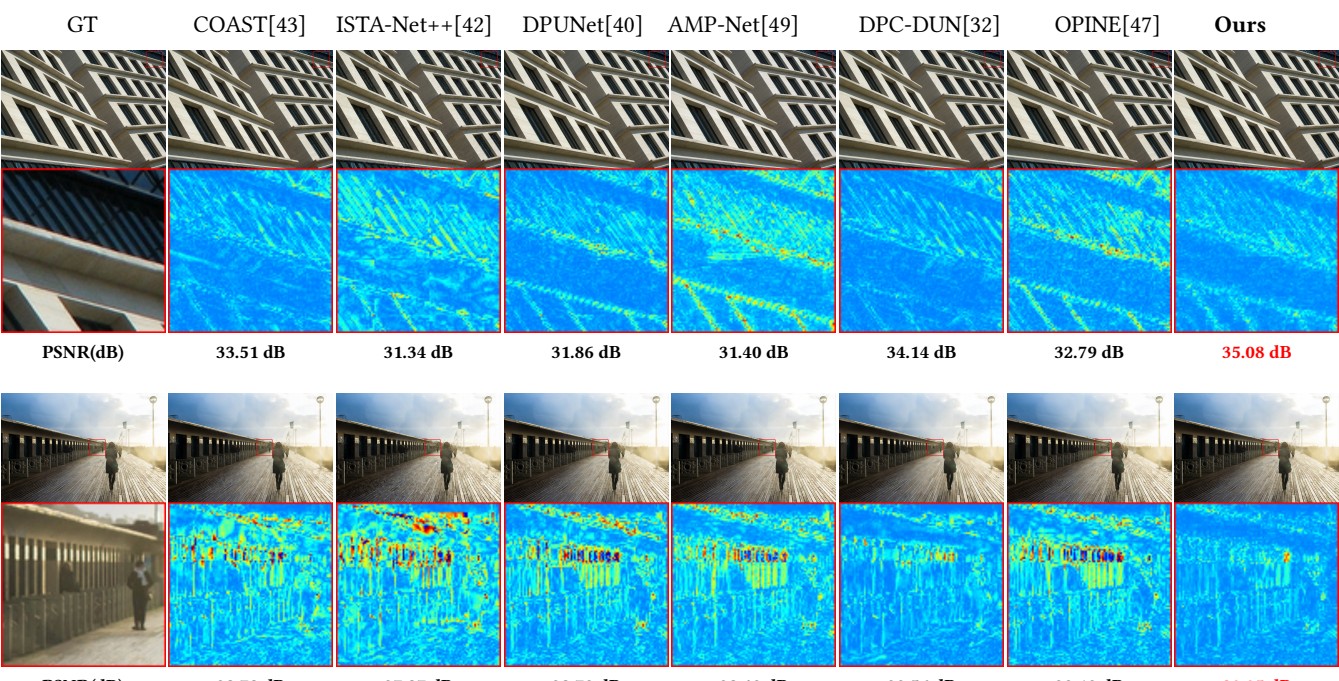

**Figure 5: Visual quality comparisons on Urban100 at 10% CS sampling ratio. We present a series of residual maps, where the color will gradually change from blue to red as the error increases, illustrating the difference between reconstructions. The best results are highlighted in red.**

the effect of inter-stage transmission of multi-scale features.

**Robustness to Noise.** To investigate the robustness of our method, we first introduced Gaussian noise with different noise levels into the CS measurements. Then, both our method and other methods utilize these noisy measurements as input to their respective reconstruction networks. Fig. 7 illustrates the PSNR/SSIM values of all methods against different standard deviations of noise on Set11 when the CS sampling ratio is 10%. The results clearly indicate that our method exhibits superior robustness to noise corruption compared to other comparing methods.

**Complexity Analysis.** Table 6 compares the parameters, reconstruction time, and average PSNR for reconstructing 256×256 images at a 50% CS sampling ratio. Owing to the superior computing

power of GPUs, the slight difference in the running time is not important, image reconstruction quality is more important for deep learning-based methods. In summary, our method achieves a better accuracy-complexity trade-off than other superior methods.

**Discussion on the Number $k$ of Stages.** As presented in Table 7, the selection of $k = 9$ for the number of stages in our model is based on a detailed analysis of reconstruction outcomes on various datasets. For brevity, we only display the test results at a 10% sampling rate on the Set11 dataset. Our model exhibits satisfactory performance when $k = 9$, suggesting that further iterations would not substantially enhance the reconstruction quality. To balance the computational efficiency and reconstruction accuracy, the ablation studies primarily employed $k = 9$, thus optimizing the performance

**Table 2: Average PSNR(dB)/SSIM comparisons with different methods on Set5 and Set11. The best and second-best results are in bold and underlined, respectively.**

| Dataset | Method | Sampling Rate | | | | |
|---|---|---|---|---|---|---|
| | | 10% | 20% | 30% | 40% | 50% |
| Set5 | AMP-Net [49] | 31.95/0.9017 | 35.49/0.9419 | 37.86/0.9606 | 39.70/0.9713 | 41.51/0.9791 |
| | COAST [43] | 30.50/0.8794 | 34.18/0.9298 | 36.48/0.9515 | 38.33/0.9645 | 40.21/0.9744 |
| | ISTA-Net [46] | 28.53/0.8277 | 32.22/0.8995 | 34.87/0.9354 | 37.00/0.9546 | 39.09/0.9684 |
| | ISTA-Net++ [42] | 29.61/0.8563 | 33.33/0.9173 | 35.62/0.9427 | 37.40/0.9575 | 38.94/0.9678 |
| | MADUN [31] | 31.11/0.8910 | 34.80/0.9363 | 37.25/0.9561 | 39.29/0.9693 | 41.18/0.9784 |
| | ULAMP [38] | 30.78/0.8774 | 31.94/0.8868 | 36.39/0.9599 | 38.20/0.9693 | 40.97/0.9827 |
| | DPUNet [40] | 31.80/0.9079 | 35.38/0.9458 | 37.54/0.9618 | 39.44/0.9716 | 41.10/0.9783 |
| | DPC-DUN [32] | 31.12/0.8927 | 34.62/0.9351 | 37.16/0.9558 | 39.14/0.9686 | 41.08/0.9779 |
| | SODAS-Net [33] | 30.59/0.8800 | 34.05/0.9288 | 36.86/0.9542 | 38.98/0.9678 | 40.87/0.9771 |
| | **Ours** | **33.19/0.9220** | **36.43/0.9530** | **38.85/0.9681** | **40.92/0.9780** | **43.22/0.9850** |
| Set11 | AMP-Net [49] | 29.46/0.8792 | 33.16/0.9325 | 35.91/0.9577 | 38.17/0.9711 | 40.22/0.9801 |
| | COAST [43] | 28.69/0.8618 | 32.54/0.9251 | 35.04/0.9501 | 37.13/0.9648 | 38.94/0.9744 |
| | ISTA-Net [46] | 26.26/0.7961 | 30.24/0.8910 | 33.08/0.9316 | 35.38/0.9532 | 37.42/0.9675 |
| | ISTA-Net++ [42] | 28.34/0.8531 | 31.66/0.9127 | 34.23/0.9427 | 36.28/0.9593 | 37.94/0.9693 |
| | MADUN [31] | 29.29/0.8768 | 33.30/0.9355 | 36.00/0.9576 | 38.09/0.9700 | 39.86/0.9774 |
| | BCS-Net [50] | 29.36/0.8650 | 32.87/0.9254 | 35.40/0.9527 | 36.52/0.9640 | 39.58/0.9734 |
| | DPUNet [40] | 29.30/0.8815 | 33.17/0.9357 | 35.75/0.9581 | 37.90/0.9705 | 39.69/0.9782 |
| | DPC-DUN [32] | 29.40/0.8798 | 33.10/0.9334 | 35.88/0.9570 | 37.98/0.9694 | 39.84/0.9778 |
| | SODAS-Net [33] | 28.84/0.8665 | 32.20/0.9243 | 35.54/0.9545 | 37.72/0.9680 | 39.59/0.9769 |
| | **Ours** | **30.78/0.8985** | **34.41/0.9453** | **37.26/0.9651** | **39.22/0.9746** | **41.16/0.9818** |

**Table 3: Ablation study on the effect of dual-domain information. Average PSNR/SSIM comparisons at 10% CS sampling ratio. The best results are highlighted in bold.**

| Case | Module | | Dataset (PSNR/SSIM) |
|---|---|---|---|
| | Image domain | Wavelet domain | Urban100 |
| (a) | ✓ | ✗ | 27.67/0.8537 |
| (b) | ✗ | ✓ | 27.62/0.8526 |
| (c) | ✓ | ✓ | **28.01/0.8611** |

**Table 4: Ablation study on the effect of consistency and difference. Average PSNR/SSIM comparisons at 10% CS sampling ratio. The best results are highlighted in bold.**

| Case | Module | | Dataset (PSNR/SSIM) |
|---|---|---|---|
| | Consistency | Difference | Urban100 |
| (a) | ✓ | ✗ | 27.61/0.8521 |
| (b) | ✗ | ✓ | 27.73/0.8553 |
| (c) | ✓ | ✓ | **28.01/0.8611** |

**Table 5: Ablation study on the effect of inter-stage transmission (IST). Average PSNR/SSIM comparisons at 10% CS sampling ratio. The best results are highlighted in bold.**

| Case | Module | Dataset (PSNR/SSIM) | |
|---|---|---|---|
| | IST | Set11 | Urban100 |
| (a) | ✗ | 30.70/0.8971 | 27.85/0.8572 |
| (b) | ✓ | **30.78/0.8985** | **28.01/0.8611** |

**Table 6: Model complexity comparison. PN and PM are the number of the learnable matrix parameters and total parameters, respectively. Time taken is computed at 50% CS sampling ratio on 256×256 images.**

| Method | Parameters | | | |
|---|---|---|---|---|
| | PN(Mb) | PM(Mb) | Times(s) | PSNR |
| ISTA-Net | 1.05 | 2.57 | 0.083 | 38.95 |
| OPINE-Net | 2.13 | 4.18 | 0.099 | 40.98 |
| AMP-Net | 2.13 | 5.40 | 0.072 | 41.27 |
| COAST | - | 8.56 | 0.093 | 39.88 |
| ISTA-Net++ | 2.13 | 5.80 | 0.082 | 38.83 |
| MADUN | 2.13 | 23.04 | 0.177 | 40.86 |
| DPUNet | - | 12.1 | 0.071 | 40.48 |
| **Ours** | 8.00 | 19.54 | 0.201 | 42.05 |

**Table 7: Average PSNR(dB)/SSIM comparisons with different stages at 10% CS sampling ratio on Set11 dataset.**

| Stage | k = 1 | k = 3 | k = 6 | k = 9 |
|---|---|---|---|---|
| PSNR(dB) | 28.94 | 29.89 | 30.43 | 30.78 |
| SSIM | 0.8704 | 0.8851 | 0.8929 | 0.8985 |

of our model within the constraints of available computational resources.

## 5 Conclusion

We propose a novel dual-domain framework that explores the direct visual representation from the image domain and multi-scale

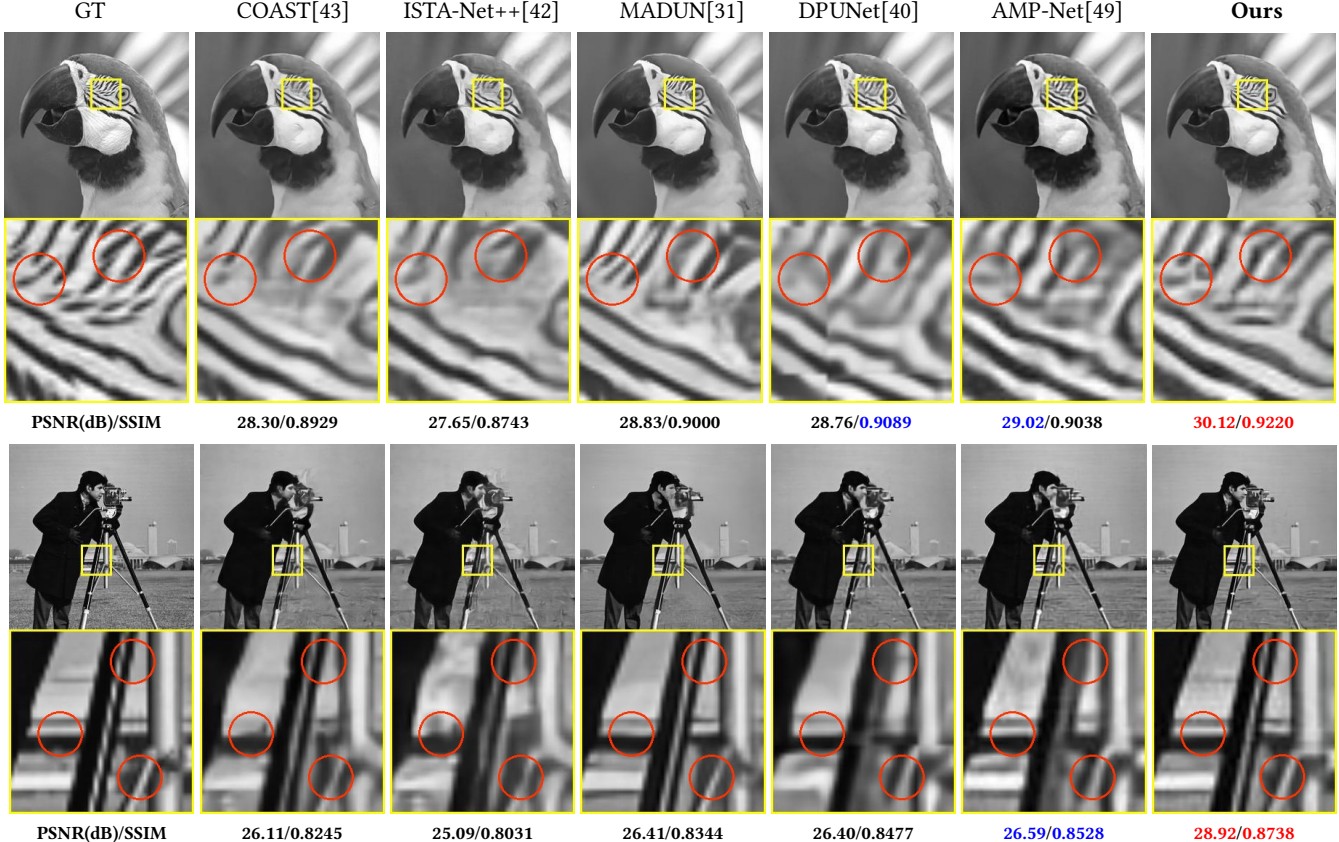

| | GT | COAST[43] | ISTA-Net++[42] | MADUN[31] | DPUNet[40] | AMP-Net[49] | **Ours** |
|---|---|---|---|---|---|---|---|
| PSNR(dB)/SSIM | | 28.30/0.8929 | 27.65/0.8743 | 28.83/0.9000 | 28.76/0.9089 | 29.02/0.9038 | 30.12/0.9220 |
| PSNR(dB)/SSIM | | 26.11/0.8245 | 25.09/0.8031 | 26.41/0.8344 | 26.40/0.8477 | 26.59/0.8528 | 28.92/0.8738 |

**Figure 6: Visual quality comparisons between our method and state-of-the-art CS methods on Set11 at 10% CS sampling ratio. The best and second-best results are highlighted in red and blue, respectively.**

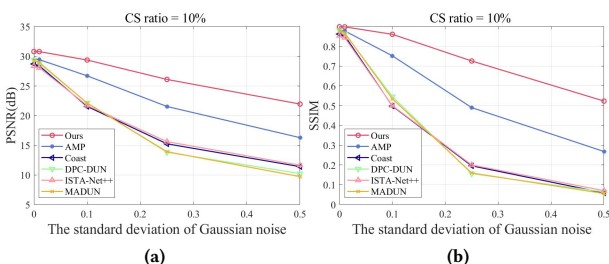

**Figure 7: Comparison of robustness to Gaussian noise on Set11 dataset at 10% CS sampling ratio. Experimental results demonstrate that the PSNR/SSIM curves of our method exhibit a notably less steep decline compared with other methods when exposed to increasing noise levels. This indicates that our method possesses a high degree of robustness against noise-induced errors. Specifically, subplot (a) presents the PSNR curve, while subplot (b) depicts the SSIM curve.**

information provided by the wavelet domain. In this paper, we overcome the challenge of information loss resulting from the inefficient representation within a single domain by employing a dual-domain collaborative optimization scheme. The integration of information from both the spatial and frequency domains allows for complementary strengths, breaking the limitations of inefficient representation in a single domain. An innovative mechanism is

designed to exploit the consistency and difference between the spatial and frequency domains. This mechanism not only utilizes consistency to guide cross-domain collaboration but also explores differences to facilitate information compensation. Finally, an inter-stage information pathway is established to efficiently broadcast multi-scale features in a frequency-adaptive manner, which mitigates the loss of intrinsic information. Extensive experimental results on various benchmark datasets demonstrate the superior performance of our method. However, a limitation of our dual-domain model arises from its heightened computational demand when handling extensive datasets and complex algorithms, particularly in the processing of high-resolution images and videos. To address this issue, we will further refine our framework to enhance its applicability across a wide range of image inverse problems and video applications, mitigating the challenges associated with large-scale data and intricate algorithmic operations.

## Acknowledgments

This work was supported by the National Key R&D Program of China (No. 2021ZD0111902), and the National Natural Science Foundation of China (62272016, 62372018, U21B2038).

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
