# OpenReview forum: "D$^3$U-Net: Dual-Domain Collaborative Optimization Deep Unfolding Network for Image Compressive Sensing"
_acmmm.org/ACMMM/2024/Conference — MM2024 Poster_

### Official Review · Reviewer_pT5H · 2024-05-23

**Rating:** 5
**Confidence:** 3

**Summary:**

This paper is well-written, and the main idea of this work is clearly presented. The authors introduce the dual-domain collaborative optimization into the deep unfolding network (DUN). This idea carries reference value for future advancements in deep unfolding techniques.

**Strengths:**

1.Novel method to adopt dual-domain collaborative optimization for image compressive sensing.

2.The introduction provides a thorough background and the methodology is presented in a clear and accessible manner.

**Limitations:**

1.Figs 1&2 are not referenced in the paper.

2.Missing some key references. Like “G$^2$-DUN: Gradient Guided Deep Unfolding Network for Image Compressive Sensing”

3.Some intuitive visual representations of features captured from dual-domain are expected to show what is learned from dual-domain optimization.

**Suitability:**

2

---

### Official Review · Reviewer_Xqmh · 2024-05-24

**Rating:** 5
**Confidence:** 3

**Summary:**

In this paper, authors propose a deep unfolding framework D3U-Net that establishes a dual-domain collaborative optimization scheme. This framework introduces both visual representations from the image domain and multi-resolution analysis provided by the wavelet domain. The proposed method design a consistency-difference collaborative mechanism to capture inter-domain dependencies effectively. This mechanism not only enhances the fidelity of reconstruction but also enriches the depth and breadth of extracted features, improving the overall robustness and reconstruction quality. Extensive experimental results on various benchmark datasets show the superior performance of our method.

**Strengths:**

1. There are three main contributions: 1) A novel dual-domain collaborative optimization framework named D3U-Net is proposed for image CS, breaking the limitation of inefficient representation within single-domain. 2) A consistency-difference collaborative mechanism is designed to capture inter-domain dependencies. 3)  An inter-stage transmission path is presented to efficiently broadcast the multi-scale features in a frequency-adaptive manner.
2. The idea of this paper sounds very interesting. Experiments demonstrate that our proposed method achieves excellent performance.
3. The figures and formulas are clear and beautiful, making them easy to read.

**Limitations:**

1. In Eq. 28, How were the coefficients 0.001 and 0.0001 of loss 2 and L3 determined? There seems to be no explanation or analysis about this part in this paper.
2. In the experimental section, there is a lack of introduction to the test dataset (Set5, Set1 and Urban100).
3. Table 6 clearly shows that the method proposed in this paper has a significantly higher number of model parameters and longer inference time compared to other methods. Specifically, compared to AMP Net, the parameters have increased approximately fourfold, and the inference time has tripled, while the PSNR has only seen a marginal increase of about 0.8. The authors argue that "Due to the superior computing power of GPUs, the slight difference in running time is not significant, as image reconstruction quality is more important for deep learning-based methods." However, I respectfully disagree with this perspective. I contend that both reconstruction quality and speed are of equal importance, as the speed of reconstruction significantly impacts the model's applicability and escalates application costs. Providing different testing platforms for different methods would foster a more equitable comparison.

**Suitability:**

2

---

### Official Review · Reviewer_fuEB · 2024-05-28

**Rating:** 3
**Confidence:** 4

**Summary:**

The paper introduces D3U-Net, a novel framework for image compressive sensing (CS) that integrates visual representations from the image domain with multi-resolution wavelet domain analysis. It proposes a dual-domain collaborative optimization framework that helps to accurately constrain the feasible solution space for image reconstruction. A unique consistency-difference collaborative mechanism within the framework captures inter-domain dependencies, enhancing the fidelity and robustness of the reconstructed images. Additionally, the paper outlines an inter-stage transmission pathway that reduces information loss and effectively broadcasts multi-scale features in a frequency-adaptive manner. Experimental results demonstrate that D3U-Net outperforms previous techniques, offering improvements in image compressive sensing technology.

**Strengths:**

The paper presents a collaborative optimization scheme that leverages the strengths of both spatial and frequency domains. By capitalizing on this duality, the framework ensures a more comprehensive and efficient image representation.

**Limitations:**

1. In the related work of this paper, the existing compressed sensing algorithms have been divided into deep black box networks and deep unfolding networks, but in experiments, different algorithms were not compared according to this classification standard.
2. For the proposed algorithm, the coco dataset is used as the training set. However, for the compared algorithms, the provided pre-trained models rarely use this dataset as the training set, which leads to unfair comparisons with other algorithms.
3. In this paper, many execution details were not clearly explained. For example, whether the proposed algorithm adopted a block-based sampling strategy and the block size setting, as well as is the sampling matrix learnable?
4. Is the learnable matrix in Table 6 the sampling matrix? Why is the PN (in Table 6) of the proposed algorithm different from other CS algorithms?
5. Is the same sampling matrix used by different algorithms in Figure 6? Or do different algorithms utilize their own sampling matrices?

**Suitability:**

3

---

### Meta-Review · Area_Chair_MJMe · 2024-07-02

**Recommendation:** Accept (Poster)
**Confidence:** 5

**Metareview:**

After rebuttal, this paper receives 2 positive reviews (Accept and Weak Accept) and 1 negative review (Borderline Reject), so I recommend the acceptance of this paper.